# Detection Method and Experimental Research of Leafy Vegetable Seedlings Transplanting Based on a Machine Vision

**Wei Fu [1], Jinqiu Gao [1], Chunjiang Zhao [2,\*], Kai Jiang [3,\*], Wengang Zheng [3] and Yanshan Tian [4]**

[1] College of Mechanical and Electrical Engineering, Hainan University, Haikou 570228, China
[2] Research Center of Information Technology, Beijing Academy of Agriculture and Forestry Sciences, Beijing 100097, China
[3] Research Center of Intelligent Equipment, Beijing Academy of Agriculture and Forestry Sciences, Beijing 100097, China
[4] School of Mathematical and Computer Science, Ningxia Normal University, Guyuan 756000, China
[\*] Correspondence: zhaocj@nercita.org.cn (C.Z.); jiangk@nercita.org.cn (K.J.); Tel.: +86-10-5150-3504 (K.J.)

**Abstract:** In view of the need to remove empty cells and unqualified seedlings for automatic transplanting of leafy vegetable seedlings, this paper proposes a method to detect the growth parameters of leafy vegetable seedlings by using machine vision technology. This method uses the image processor PV200 to perform image grayscale, threshold segmentation, corrosion, expansion, area division, etc. to obtain the pixel value of the leaf area of the seedling and compare it with the set standard value, which provides guiding information for eliminating empty cells and unqualified seedlings. Lettuce seedlings at 17 days, 20 days, and 22 days of seedling age were used as the test objects, and the growth status and test results of the seedlings were analyzed to determine the optimum seedling age for transplanting. The test results show that there is basically no leaf cross-border between the lettuce seedlings at the age of 17 days, the average pixel area of the leaves is 3771.74, and the detection accuracy rate is 100%; the seedlings at the age of 22 days grow 5–6 leaves, the detection accuracy of unqualified seedlings and qualified seedlings was 62.50% and 88.16%, respectively, and the comprehensive detection accuracy was 85.71%. The comprehensive detection accuracy rate showed a downward trend with the increase of seedling age, mainly due to the partial occlusion between leaves. The transplanting of leafy vegetable seedlings is a sparse transplanting operation, and the seedling spacing increases after transplanting. Therefore, the detection of seedlings in the process of transplanting can greatly improve the recognition accuracy and solve the problem that the leaves of the seedlings in the seedling tray are obscured by each other and affect the detection accuracy. The research results can provide a theoretical basis and design reference for the development of the visual inspection system and the transplanting actuator of the leafy vegetable seedlings transplanting robot.

**Keywords:** leafy vegetable seedlings; transplanting robot; visual image; leaf area pixels; transplanting actuator

## 1. Introduction

China's facility horticulture industry has been developing rapidly, and the demand for intelligent equipment is extremely urgent [1,2]. Facility horticulture production robots are a current trend in the research and development of global agricultural machinery and equipment technology [3–5]. Hydroponic technology in facility horticulture can accurately supply nutrient solution and intelligently control environmental factors according to the growth needs of leafy vegetables, which has advantages of high level of production safety, being unaffected by seasons, being easy to control, etc., and is widely used [6–8]. In the production of hydroponic leafy vegetables, the transplanting of seedlings is labor-intensive and low in efficiency. Manual transplanting relies on production experience and

the naked eye to determine whether the seedlings meet the standards for transplanting, and cannot meet the requirements of factory and standardized production. Automatic sorting and transplanting is the key to improving the production efficiency of leafy vegetables in factories [9–11]. However, due to the effects of seed emergence rate, missed sowing of sowing equipment, and seedling raising environment, empty cells or unqualified seedlings appear, which affects the quality of seedling cultivation [12]. In order to realize the automatic transplanting operation, it is necessary to establish a seedling transplanting standard. In addition, before transplanting, it is also necessary to remove the empty cells and unqualified seedlings and supplement with the qualified seedlings, which can help realize the efficient operation of automatic transplanting.

Research institutions in China and abroad have carried out a lot of research on seedling detection technology and transplanting equipment. Iron OX Corporation of the United States [13] developed a robotic vision system for leafy vegetable seedling transplanting, which uses binocular recognition technology to detect individual seedlings. The detection accuracy of empty cells and seedling leaf area is high, but the operation efficiency is low. Visser, a company of the Netherlands [14], developed a visual inspection system for transplanting flower seedlings, which has a high success rate in detecting whole seedlings with small seedlings, and has been applied in production. In terms of seedling image detection methods, Ling et al. [15] adopted a multi-modal segmentation algorithm to segment leaves, the measurement error of leaf area is 2.6–2.3%. Onyango et al. [16] developed a segmentation algorithm that combines color and hole features, and the detection accuracy rate is 82–96%. Jiang et al. [17] used image processing technology to calculate the leaf area, number of leaves, perimeter, and other information of seedlings to determine the growth status of seedlings, so as to provide reference for transplanting and thinning, and the detection accuracy of tomato seedlings is 98%. Tong et al. [18] proposed a cross-border leaf region center decision method and an overlapping leaf image segmentation method, with a detection accuracy of 98.7%. Zhang et al. [19] used image processing software (CKVisionBuilder) to detect plug lettuce seedlings aged 13 days, and the accuracy rate of empty cell detection was 95.8%. Ryu et al. [20] developed a machine vision system for plug tray seedling transplanting, with an accuracy rate of 95% for cavity detection. Feng et al. [21] designed a vision system based on linear structured light, which can obtain the leaf area and plant height of cucumber seedlings in real time, and the height measurement accuracy of seedlings under the condition that the leaves do not overlap was ±5 mm. Hu et al. [22] used the least squares method to linearly fit the leaf area of tomato seedlings, and the relative error of measurement was less than 1.4%. Yang et al. [23] used visual algorithms to extract the image of the seedling stem and the projected area of the seedling stem to determine the quality of the seedling, with a detection accuracy of 97.92%, and the center position of the seedlings in the hole and the feeding speed of the plug had a great influence on the identification accuracy. In terms of quality detection of seedlings in field transplanters, Jin et al. [24,25] proposed a fuzzy C-mean detection method and an edge detection algorithm for seedlings, optimized the path of the transplanting end effector to take seedlings, and detection accuracy rates of empty cells, qualified seedlings, and unqualified seedlings were 96.42%, 98.77%, and 89.95%, respectively. Zhang et al. [26] used the maxIOU algorithm to calculate the image segmentation threshold, and the detection accuracy was 97.4%. Wen et al. [27] developed a field transplanter with the functions of seedling detection, seedling removal, and seedling replenishment, and the detection accuracy was 96.99%.

The above studies are mainly aimed at the detection of young seedlings in plug trays, and the identification of large seedlings when leaves are obscured and crossed has not yet been solved. This paper proposes to measure the leaf area parameters of lettuce seedlings of different seedling ages based on machine vision detection technology and study the image detection and processing methods. By building an image acquisition and testing platform, we obtained and detected the images of lettuce seedlings, analyzed the leaf growth status and detection accuracy of lettuce seedlings of different seedling ages, and

determined the appropriate transplanting seedling age and the design scheme of the transplanting actuator. The research results can provide reference for the development of leafy vegetable transplanting robots.

## 2. Materials and Methods

### 2.1. Hardware System

The vision system is used to provide information on empty cells, unqualified seedlings, and qualified seedlings for the transplanting operation of the leafy vegetable seedling transplanting robot, which is convenient for subsequent removal or replenishment of seedlings. Its structure includes seedling tray conveyor, transplanting plate conveyor, handling robot, transplanting end effector, vision system, and control cabinet, as shown in Figure 1. The position relationship between the robot and the two conveyor belts is for taking and transplanting seedlings. The photoelectric sensor is used to detect the signals of seedling tray and transplant board, and then the mechanical parts are used for secondary positioning. In the robot control system, the space coordinates of taking, detection, and transplanting seedlings are set. The position error of robot movement is ± 0.02 mm.

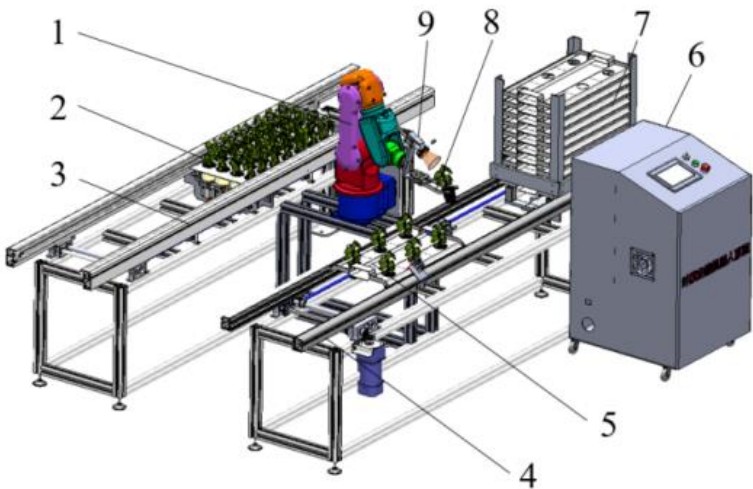

**Figure 1.** Leafy vegetable transplantation robot: 1. Handling robot. 2. Seedling tray. 3. Seedling tray conveyor. 4. Transplanting plate conveyor. 5. Transplanting plate. 6. Control cabinet. 7. Transplanting plate lower plate mechanism. 8. Transplanting actuator. 9. Vision system.

Working process: First, start the seedling tray conveyor and the transplanting plate conveyor, and send the seedling tray and transplanting plate to the seedling picking station and the transplanting station, respectively. The handling robot moves to the seedling picking station and drives the transplanting actuator to grab the transplanting cup. Then, the handling robot transports the transplanting cup to the detection station, and the vision system analyzes and calculates the pixel value of the seedling leaf area and compares it with the set value. If the detection value is greater than or equal to the set value, the vision system sends a qualified seedling signal to the robot. The handling robot drives the transplanting actuator to put the transplanting cup into the transplanting plate. If the detected value is less than the set value, the vision system sends a signal of unqualified seedlings to the robot, and the handling robot discards the unqualified seedlings. Then, the transplanting robot takes seedlings, detects and conducts transplanting again, and repeats the cycle. The vision system includes a stand, a camera, a controller, operating handle, and display, as shown in Figure 2. The system employs Panasonic ANPVC2260 CCD color camera (effective pixel 2 million), PV200 vision controller, ANPVP03 operating handle (Panasonic, Toranomon 35 Mori Building, 3-4-10 Toranomon Minato-ku, Tokyo 105-0001 Japan), and DF6HA-1S High-definition wide-angle lens (Chongqing Judian Technology Co., Ltd., 14-1, Building C, West Mall, No. 15 Shiyang Road, Jiulongpo District,

Chongqing, China). Fixed on the bracket, the camera is 465 mm away from the upper surface of the seedling tray. The camera, the operating handle, and the display are connected to each interface of the controller through a data cable. The system uses the operating handle to call the internal image processing program of the PV200 controller to identify and analyze the seedling leaf area, center coordinates, detection area, and other indicators.

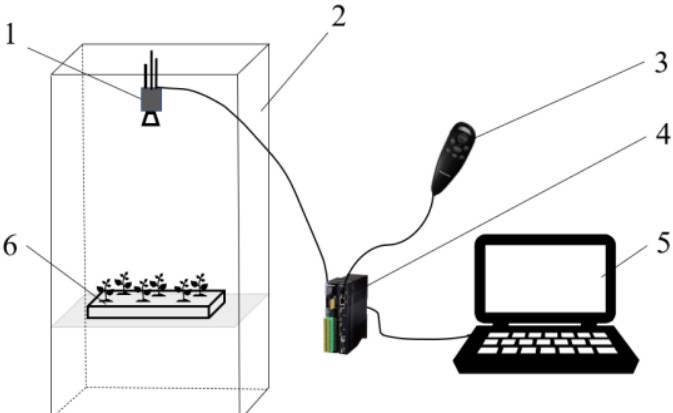

**Figure 2.** The composition of vision system: 1. Camera. 2. Stand. 3. Handle. 4. Controller. 5. Monitor. 6. Seedling tray.

### 2.2. Software System

The PV200 vision controller has built-in image processing software PVWIN200. By calling image processing instructions to write detection programs, PVWIN200 conducts collection, segmentation, corrosion, expansion, region specification, pixel counting, numerical operation, and judgment output over the images of leafy vegetable seedlings. The system achieves rapid detection of seedling growth indicators and transmits the position and growth status of the seedlings to the robot control system through the communication between the PV200 vision controller and the transplanting robot, which provides the basis for the operation of removing seedlings, replenishing seedlings and transplanting.

### 2.3. Image Detection Method

The image detection processing flow is shown in Figure 3. First, the system reads the color image taken by the camera, changes the grayscale of the image to obtain a grayscale image, then performs threshold segmentation on the grayscale image to obtain a binary image, erodes and dilates the binary image to remove noise interference points, and finally divides the detection area and counts the pixel values in each detection area.

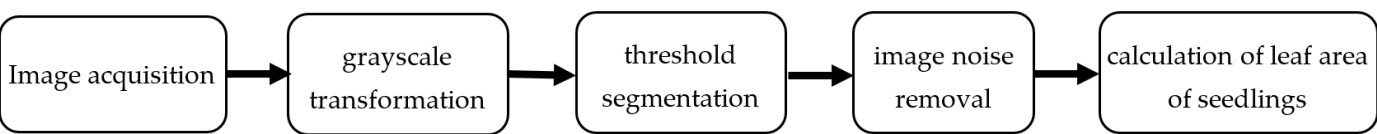

**Figure 3.** Flowchart for image processing.

### 2.3.1. Grayscale Image

The RGB color model is a color model represented by a unit cube in the spatial Cartesian coordinate system, where R, G, and B represent the red, green, and blue color components of each pixel, respectively. In order to distinguish green plants from the background, the grayscale factor increases the G value component and reduces the R and B value components. The grayscale factor 2G-R-B is selected for grayscale processing, see Formula 1. This factor has a small amount of calculation and a good effect of capturing

"green" [28–30]. The processing result is shown in Figure 4a. It can be seen from Figure 4a that the seedling tray and the transplanting cup are clearly visible, and the grayscale effect is not good, which is not conducive to the subsequent threshold segmentation. This is because the images of the seedling tray and the transplanting cup are both white and contain a part of the green component. After many experiments, the optimal grayscale factor G+R-2B is obtained, see Formula 2. It can be seen from Figure 4b that the separation effect of green plants and the background is good, but because the gray value of the plant leaves is small and the contrast with the background is small, it is also not conducive to the subsequent threshold segmentation. The automatic tone correction method is used to adjust the contrast of Figure 4b, and the processing result is shown in Figure 4c. The difference between the plants and the background is obvious, which is beneficial to the subsequent image segmentation processing.

$$f(x, y) = \begin{cases} 0, & 2G - R - B < 0 \\ 2G - R - B, & 2G - R - B \geq 0 \end{cases} \quad (1)$$

$$f(x, y) = \begin{cases} 0, & G + R - 2B < 0 \\ G + R - 2B, & G + R - 2B \geq 0 \end{cases} \quad (2)$$

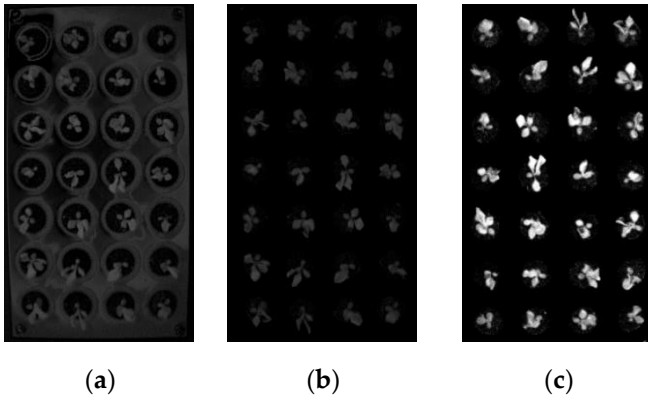

(**a**)  (**b**)  (**c**)

**Figure 4.** Gray image of lettuce plug seedling with seedling age of 17d. (**a**) 2G-R-B; (**b**) G+R-2B; (**c**) automatic tone correction.

### 2.3.2. Threshold Segmentation

Threshold segmentation usually adopts ordinary threshold segmentation and Otsu algorithm. The Otsu algorithm traverses all possible thresholds and calculates the variance of two types of pixels (i.e., lower than the threshold and higher than the threshold) of each threshold result, and the operation efficiency is relatively low. According to the design requirements of leafy vegetable seedlings' environmental background, transplanting mode, and work efficiency, we selected a common threshold segmentation method with simple calculation and high reliability. Then we observed the grayscale histogram of the image and selected an appropriate threshold to binarize the grayscale image. Through repeated tests, the image segmentation effect is the best when the threshold is 60. At this time, the gray value of the lettuce is 255, and the gray value of the background is 0, as shown in Figure 5.

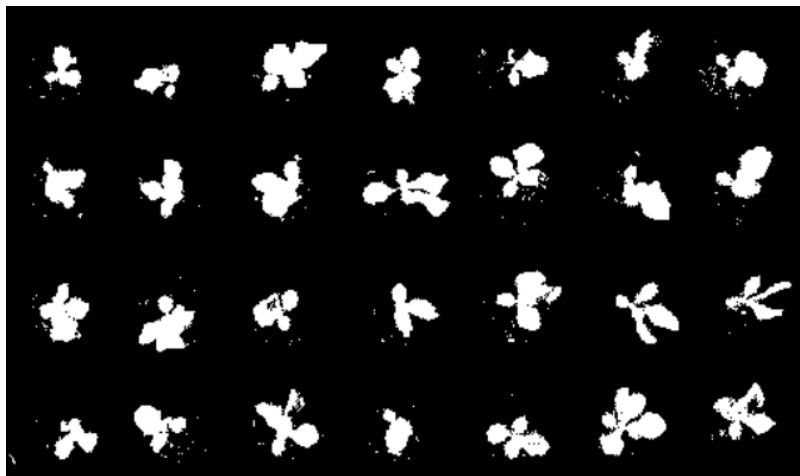

**Figure 5.** Binary image of lettuce plug seedling with seedling age of 17 d.

### 2.3.3. Corrosion and Expansion

The image after threshold segmentation contains noise, which has a great impact on the accuracy of subsequent processing such as the calculation of seedling leaf area. Removing noise is a necessary part of image processing. Observing Figure 5, it can be seen that compared with the leaf area of seedlings, the noise is mostly a single connected area with a small area. In order to eliminate the noise and not change the pixel value of the leaf area of the seedling, the morphological opening operation method (corrosion first and then expansion) is used. First, the image is eroded with 5 × 5 unit matrix structural elements to remove noise, and then the image is expanded with 5 × 5 unit matrix structural elements to restore the original image pixels. The processing result is shown in Figure 6. The method basically eliminates the noise, and the image extraction effect is good.

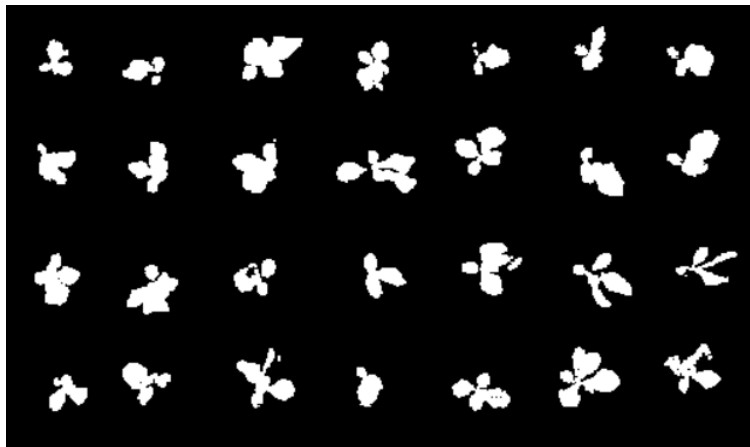

**Figure 6.** Binary image after de-noising of lettuce plug seedling with seedling age of 17 d.

### 2.3.4. Locale

In order to obtain the image information of seedlings in the largest range, we set the outer square of the transplanting cup as the detection area of a single seedling and calibrate the camera. The proportional relationship between the actual length and the pixel length is shown in Formula 3.

$$\text{scale coefficient } (\theta) = \text{measured value } (\lambda)/\text{pixel value } (\eta) \tag{3}$$

The diameter of the known transplanting cup is 65 mm, the measured pixel value is 179.856 (rounded to 180), and the scale coefficient is 0.36. The pixel value of the side length of the square detection area of seedlings is 180, and the detection area of each seedling in

the seedling tray is divided, as shown in Figure 7. The pixel value in the detection area is the top-view leaf area of each seedling.

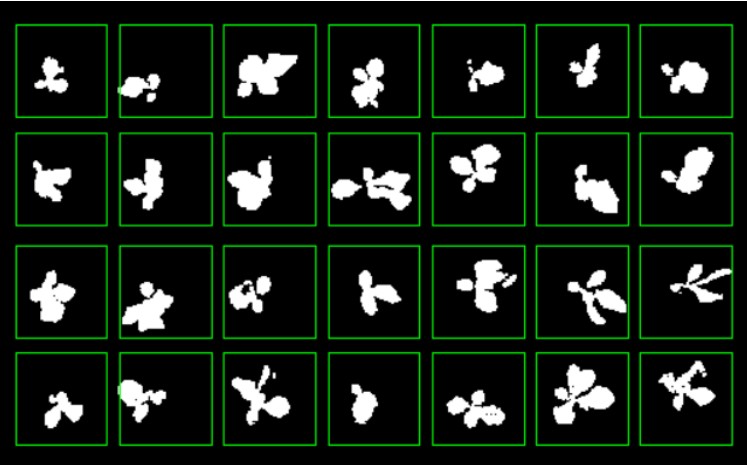

**Figure 7.** Image after area division.

### 2.3.5. Judgment and Output

After the seedling binary image is de-noised, the white part is the seedling with a pixel value of 1, and the black part is the background with a pixel value of 0. According to the area divided in Figure 7, the system counts the number of pixels in the white part of each area, and compares it with the minimum number of pixels of qualified seedlings N. When the number of pixels in the leaf area of the seedling is greater than or equal to N, it is judged as a qualified seedling, and it is marked with a green frame; otherwise, it is judged as an unqualified seedling or empty cell, which is marked with red boxes, as shown in Figure 8. When the results are output, the results of qualified seedlings are converted into a digital quantity "1", and the results of unqualified seedlings or empty cells are converted into a digital quantity "0". The judgment results are transmitted to the robot control system through serial communication to provide visual information guidance for subsequent operations such as seedling removal, seedling replenishment, and sorting.

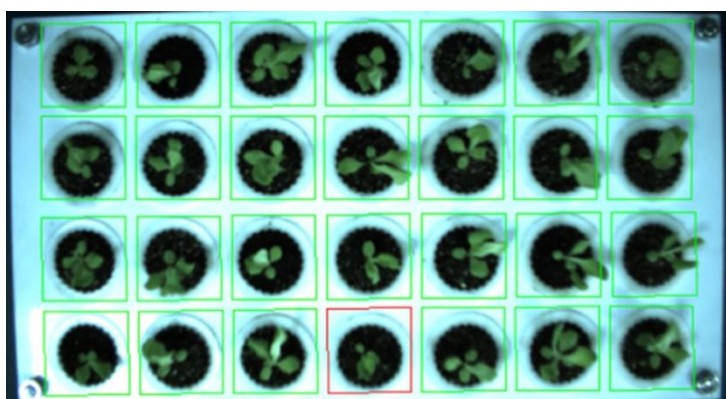

**Figure 8.** Detection results.

### 2.4. Test Content

The seedling raising test was carried out on 18 June 2022 in the solar greenhouse of Beijing Academy of Agriculture and Forestry. The substrate for seedlings was peat and vermiculite 5:1 and water was evenly mixed to achieve a water content of 40% suitable for sowing, and plastic transplanting cups are used for seedlings. The specifications of the seedling tray were 548 mm in length, 300 mm in width, 28 hole, and 72 mm in distance between the center of two adjacent holes. Seedling raising process: first, put the substrate

into the transplanting cup, artificially sow 200 lettuce seeds (Flandria RZ) in the center of the transplanting cup, cover with the vermiculite with a thickness of about 5 mm, water it thoroughly, put the transplanting cup into the tray, and build it on the greenhouse seed-bed for cultivation. The whole day temperature in the greenhouse is 19.82–40.99 °C, the average temperature is 30.75 °C; the relative humidity is 19.86–85.04%, and the average humidity is 56.03%. The test device includes a camera, a controller, a seedling tray, an operating handle, transplanting cups, a monitor, and a bracket, as shown in Figure 9.

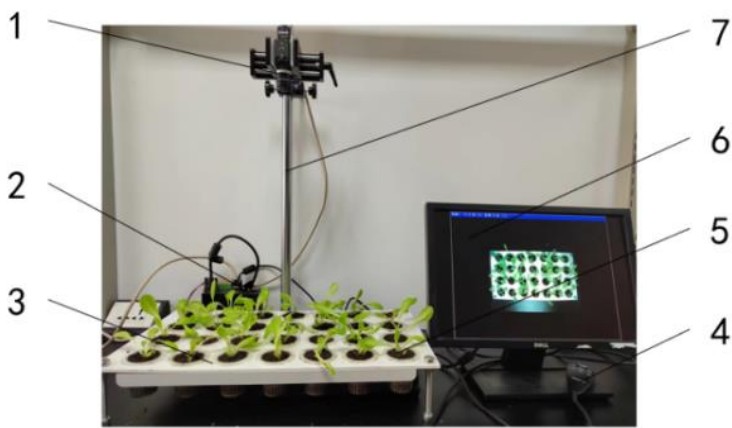

**Figure 9.** Test equipment: 1. Camera. 2. Controller. 3. Seedling tray. 4. Operating handle. 5. Transplanting cup. 6. Monitor. 7. Stand.

A total of 84 lettuce seedlings were randomly selected and placed in the seedling tray, and images were collected in the B710 laboratory of Beijing Agricultural Science Building (without supplementary light). Using DLY-1802 light meter (Delixi Group Co., Ltd., Delixi Building, No. 1 Liuqing Road, Liushi Town, Yueqing City, Zhejiang Province, China), the light intensities of the four corners of the seedling tray were measured as 131.5lx, 125.0lx, 105.2lx, and 119.1lx. When the lettuce seedlings were cultivated to the 17th day, the number of leaves was 3–4, and there was no cross-border between the leaves, which met the requirements of mechanical transplanting; when the seedling age was 20 days, some cross-border between the leaves occurred; when the seedling age was 22 days, the number of leaves was 5–6, and obvious cross-border between the leaves occurred. Therefore, lettuce seedlings aged 17 d, 20 d, and 22 d were selected for image acquisition and detection experiments to determine the appropriate transplanting age and transplanting operation plan.

## 3. Results and Discussion

### 3.1. Test Results

The color images collected at 17 d, 20 d, and 22 d of seedling age were processed by grayscale, threshold segmentation, erosion expansion, region division, and pixel counting according to the above-mentioned image detection methods. When the seedling age was 17 days, the number of pixels of the qualified seedling leaf area was set to 2500, and the detection area was judged regarding whether there was a lack of seedlings. The results are shown in Table 1.

**Table 1.** Number of pixels of leaf area of lettuce seedlings at the age of 17 days.

| Row No. | Column No. | | | | | | |
|---|---|---|---|---|---|---|---|
| | 1 | 2 | 3 | 4 | 5 | 6 | 7 |
| 1 | 2019 | 2376 | 4443 | 5396 | 4120 | 2620 | 3043 |
| 2 | 1889 | 4877 | 7218 | 4013 | 2102 | 3312 | 4383 |
| 3 | 2815 | 4408 | 4474 | 3638 | 4667 | 2163 | 2489 |

| 4 | 3644 | 2184 | 4400 | 3450 | 2407 | 2134 | 2032 |
| 5 | 3122 | 3533 | 3806 | 2916 | 5040 | 2199 | 5281 |
| 6 | 3689 | 3350 | 4855 | 2555 | 3525 | 3724 | 5073 |
| 7 | 3741 | 4624 | 3604 | 4817 | 4975 | 3641 | 3049 |
| 8 | 2480 | 4124 | 3851 | 5835 | 3451 | 3872 | 3209 |
| 9 | 2375 | 2470 | 6226 | 3890 | 2335 | 2464 | 3642 |
| 10 | 3341 | 3835 | 5111 | 5903 | 4765 | 4254 | 4503 |
| 11 | 4609 | 5015 | 3025 | 3438 | 5222 | 4096 | 2854 |
| 12 | 2817 | 3606 | 4873 | 2479 | 4131 | 6755 | 4135 |

It can be seen from Table 1 that the pixel number of the leaf area of 84 seedlings was 1889–7218, and the average pixel value was 3771.74. Among them, for the row and column coordinates (1, 1), (1, 2), (2, 1), (2, 5), (3, 6), (3, 7), (4, 2), (4, 5), (4, 6), (4, 7), (5, 6), (8, 1), (9,1), (9,2), (9,5), (9,6), (12,4), the number of pixels was less than 2500, and they were judged as unqualified seedlings; the number of pixels in the other detection areas was greater than or equal to 2500, judged as qualified seedlings. When the seedling age was 20 d, the number of pixels of qualified seedling leaf area was set to 4500, the pixel number of leaf area of 84 seedlings was 2929–9219, the average pixel value was 5697.04, and the number of unqualified seedlings was 24. When the seedling age was 22 days, the pixel number of qualified seedlings was set to 4800, the pixel number of leaf area was 2029–12447, the average pixel value was 6743.83, and the number of unqualified seedlings was 14. The test results are shown in Figure 10 where qualified seedlings are marked with green boxes and unqualified seedlings or empty cells are marked with red boxes.

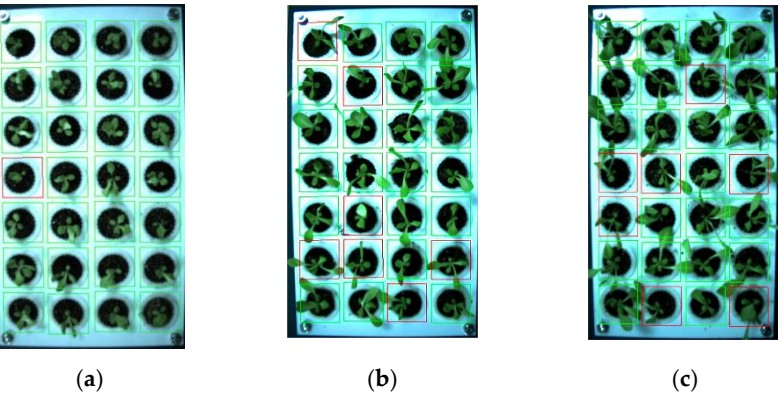

(**a**)          (**b**)          (**c**)

**Figure 10.** Detection results of lettuce seedling with different seedling age: (**a**) 17 d of seedling age; (**b**) 20 d of seedling age; (**c**) 22 d of seedling age.

The changes in the number of pixels in the leaf area of lettuce seedlings from 17 d to 22 d of seedling age were counted, as shown in Figure 11. The black line is the line connecting the pixel points of the leaf area, and the red line is the dividing line between qualified seedlings and unqualified seedlings, that is, the detection results at the lower part of the red line are unqualified seedlings. The distribution of pixel points is uneven, and the difference between the maximum value and the minimum value is obvious, indicating that the growth of leafy vegetable seedlings is quite different.

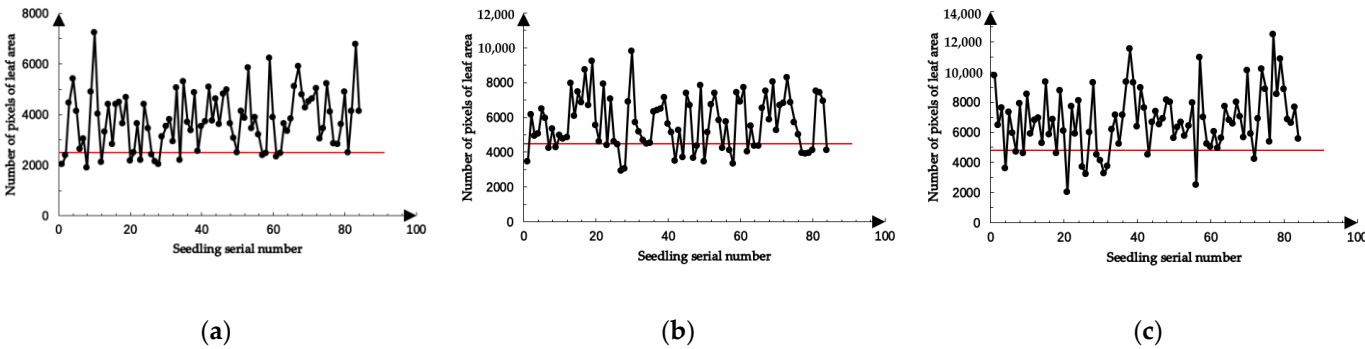

**Figure 11.** Lettuce leaf area of different seedling ages: (**a**) Seedlings at age 17 d; (**b**) seedlings at age 20 d; (**c**) seedlings at age 22 d.

The detection results of lettuce seedlings of different seedling ages were compared, the misjudgment of the vision system was analyzed, and the accuracy rate of leaf area recognition of lettuce seedlings was calculated. Misjudgments include:

1. The empty cell is judged as an unqualified seedling, which is caused by a small number of leaves blocking the empty cell in the neighboring detection area;
2. The unqualified seedling is judged as an empty cell, which occurs when the seedling is particularly small or when the position of the seedling is at the edge of the detection area, and most of the leaves grow to the adjacent detection area;
3. The empty cell is judged as a qualified seedling, which happens when a large number of leaves is in the detection area adjacent to the empty cell block the empty cell;
4. Qualified seedlings are judged as empty cells, which happens when qualified seedlings are located at the edge of the detection area, and most of the leaves grow to the adjacent detection area;
5. Unqualified seedlings are judged as qualified seedlings, which happens when the unqualified seedlings adjacent to the detection area block the unqualified seedlings without leaves or two unqualified seedlings grow into the same detection area;
6. Qualified seedlings are judged as unqualified seedlings, because the leaves of the seedlings near the edge of the detection area grow to the adjacent detection area.

Among them, conditions 1 and 2 did not affect the accuracy of detection and the conditions 3 and 4 did not occur in this experiment. The detection accuracy of the vision system is mainly affected by the situations 5 and 6.

The calculation formula of the detection accuracy of various situations is as follows:

Detection accuracy rate of unqualified seedlings = number of unqualified seedlings detected/total number of unqualified seedlings (4)

Detection accuracy rate of qualified seedlings = number of qualified seedlings detected/total number of qualified seedlings (5)

Comprehensive detection accuracy rate = 1 − (the number of qualified seedlings detected as unqualified seedlings + the number of unqualified seedlings detected as qualified seedlings)/total number of seedlings (6)

The statistical results of the detection accuracy are shown in Table 2.

**Table 2.** Statistics of detection accuracy.

| Seedling Age (d) | The Number of Unqualified Seedlings (Plants) | The Number of Unqualified Seedlings Detected (Plants) | The Number of Qualified Seedlings (Plants) | The Number of Qualified Seedlings Detected (Plants) | Detection Accuracy Rate of Unqualified Seedlings (%) | Detection Accuracy Rate of Qualified Seedlings (%) | Comprehensive Detection Accuracy Rate (%) | Cross-Border Leaves |
|---|---|---|---|---|---|---|---|---|
| 17 | 17 | 17 | 67 | 67 | 100.00 | 100.00 | 100.00 | none |
| 20 | 17 | 17 | 67 | 60 | 100.00 | 89.55 | 91.67 | a few |
| 22 | 8 | 5 | 76 | 67 | 62.50 | 88.16 | 85.71 | many |

*3.2. Discussion*

3.2.1. The Effect of Seedling Age on Detection Accuracy

It can be seen from Table 2 that the leaves of lettuce seedlings with a seedling age of 17 days basically did not cross the boundary, and the detection accuracy of unqualified seedlings, qualified seedlings, and comprehensive detection accuracy all reached 100%, indicating that the detection accuracy of leaf growth in the detection area of the transplanting cup is high. The leaves of lettuce seedlings aged 20 days crossed the boundary less, and the total number of unqualified seedlings was 17, all of which were detected, and the detection accuracy rate was 100%, indicating that less cross-border between leaves had little impact on the detection accuracy of unqualified seedlings. Among 67 qualified seedlings, 60 were detected, and the detection accuracy of qualified seedlings was 89.55%, which was 10.45% lower than the accuracy rate of 17 d seedlings. The loss of leaf area pixels caused errors in the detection results, indicating that cross-border leaves had a great impact on the detection accuracy of qualified seedlings, and the comprehensive detection accuracy dropped to 91.67%. The leaves of lettuce seedlings aged 22 days crossed the boundary more seriously. There were 8 unqualified seedlings in total, 5 were detected, and the detection accuracy was 62.50%, which was significantly lower than that of unqualified seedlings at 20 days. This is because the leaves of the adjacent detection area grew into the unqualified seedling area, resulting in the unqualified seedlings being mistakenly detected as qualified seedlings. The total number of qualified seedlings was 76, and 67 were detected, with an accuracy rate of 88.16%, a decrease of 1.39% compared with the 20-day seedling age. Fewer cross-border cases had little effect on the detection results of qualified seedlings, and the comprehensive detection accuracy dropped to 85.71%.

In addition, with the increase of seedling age, the detection accuracy rate of qualified seedlings decreased significantly, and the detection accuracy of unqualified seedlings remained basically unchanged. This is due to the fact that the growth of seedling leaves crossed the boundary, resulting in errors in the detection of some qualified seedlings, but only a very small part of qualified seedlings that crossed the boundary entered the adjacent detection area, which had little influence on the detection and judgment of unqualified seedlings. As seedling age continued to increase, the detection accuracy of qualified seedlings did not change much, and the detection accuracy of unqualified seedlings dropped sharply. It is because the seedlings crossed the boundary seriously, the number of qualified seedlings was large, and there was a situation that they crossed each other's boundaries, so the detection accuracy of qualified seedlings decreased compared with the 20 d seedling age. However, a large number of leaves from adjacent detection area entered the detection area of unqualified seedlings, resulting in a great decrease in the detection accuracy of unqualified seedlings. If the seedling age continued to increase, the detection accuracy of unqualified seedlings would gradually drop to 0%, and when the detection accuracy of qualified seedlings dropped to a certain value, it would continue to rise to 100%.

According to Table 2, it can be seen that the older the seedling age, the more serious the leaf crossing, and the lower the detection accuracy. Therefore, from the perspective of detection accuracy, the optimal transplanting time is 17–20 d. At this time, the detection accuracy rate is 91.67–100%. The detection accuracy of unqualified seedlings is 100%, and the detection accuracy of qualified seedlings is 89.55–100%. All unqualified seedlings can be eliminated, but a small number of healthy seedlings will be eliminated. From an agronomic point of view, lettuce seedlings grow to 5–6 leaves and are suitable for transplanting, that is, the seedling age is 22 days. The detection accuracy rate of unqualified seedlings is 62.50%, the accuracy rate of qualified seedlings is 88.16%, and the comprehensive detection accuracy rate is only 85.71%, which will cause a large number of unqualified seedlings not to be removed and a small number of qualified seedlings to be removed, and the comprehensive detection accuracy will be low.

### 3.2.2. Transplanting Actuator Scheme Design

The specifications of the seedling tray and the transplanting plate are shown in Figure 12. The seedling tray has 24 holes in 6 rows and 4 columns, and the seedling spacing is 72 mm; the transplanting plate has 6 holes in 3 rows and 2 columns; and the seedling spacing is 180 mm. Transplanting is to take out the transplanting cup from the seedling tray and put it into the transplanting plate, which is a sparse transplanting operation.

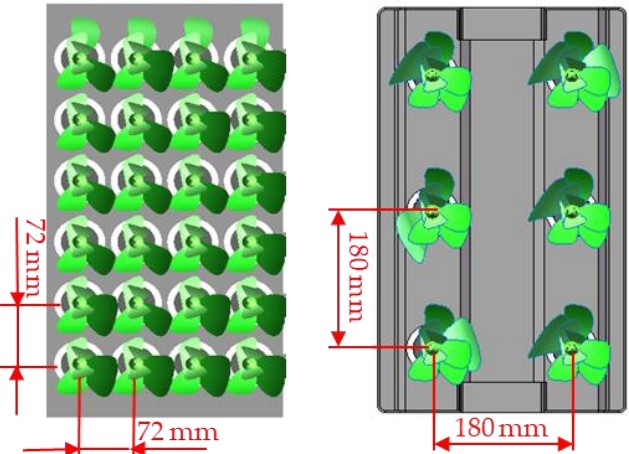

**Figure 12.** A seedling tray and a transplanting plate.

In order to shorten the cultivation time of leafy vegetable seedlings after transplanting, lettuce seedlings with a seedling age of 22 d were selected as the transplanting objects. At this time, the leaves in the seedling tray were more seriously out of bounds, and the detection accuracy was low. In order to solve the above problems, the transplanting actuator is designed in two modes: single-claw and multi-claw. The single-claw transplanting actuator is shown in Figure 13. The camera is integrated with the pneumatic hand and the gripper finger through the connecting plate, and the gripper is installed on the pneumatic finger, which can simultaneously grasp a single transplanting cup and identify seedlings.

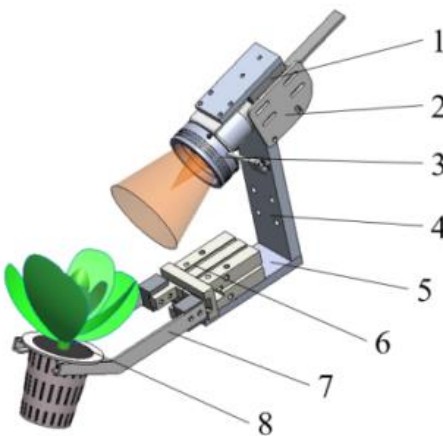

**Figure 13.** Single-claw transplanting actuator: 1. Camera. 2. Fixing plate. 3. Light source. 4. Connecting plate. 5. Cylinder fixing plate. 6. Pneumatic finger. 7. Gripper. 8. Transplanting cup.

The operation plan of the single-claw transplanting actuator is shown in Figure 14. The transplanting actuator grabs the transplanting cup in the seedling tray and transports it to the inspection station to judge whether the seedling meets the transplanting standard. If the seedling is qualified, the transplanting actuator puts the transplanting cup into the transplanting plate. If the seedlings are hollow or unqualified seedlings, discard the seedlings and re-take the seedlings, detect and transplant them, and repeat the cycle in turn. This scheme has high detection accuracy for single seedlings, but low transplanting efficiency.

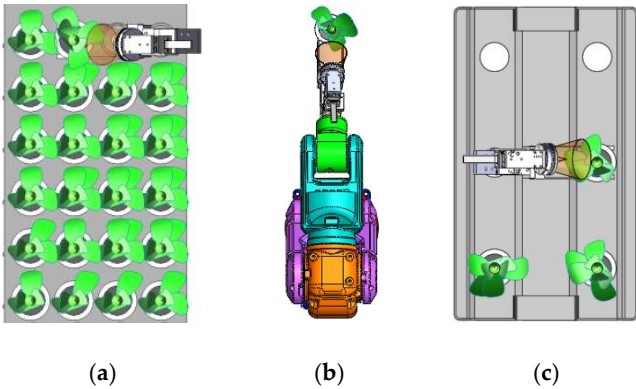

(**a**)          (**b**)          (**c**)

**Figure 14.** Single-claw transplanting actuator work plan. (**a**) Taking seedlings; (**b**) detection; (**c**) transplanting.

According to the specifications of the seedling tray and the transplanting plate, a row of 3 seedlings can be transplanted on the transplanting plate at most. The center distance of the seedlings from grabbing to transplanting is increased from 72 mm to 180 mm. Therefore, the actuator should have the function of variable spacing, thus a three-claw transplanting actuator was designed, as shown in Figure 15. The structure and driving method of the clamping jaw are the same as those of the single-claw transplanting actuator. The guide rail is installed on the main board, and the timing belt and pulley are fixed on the guide rail. Two sets of sliding blocks are installed inside the guide rails in a sliding fit and are installed and fixed with the upper and lower sides of the synchronous belt, respectively. The middle jaw is fixed in the center of the main board, and the other two sets of jaws are respectively installed on the left and right sliders. The stepper motor is used to drive the synchronous belt to rotate, and the relative movement of the left and right sliders is realized.

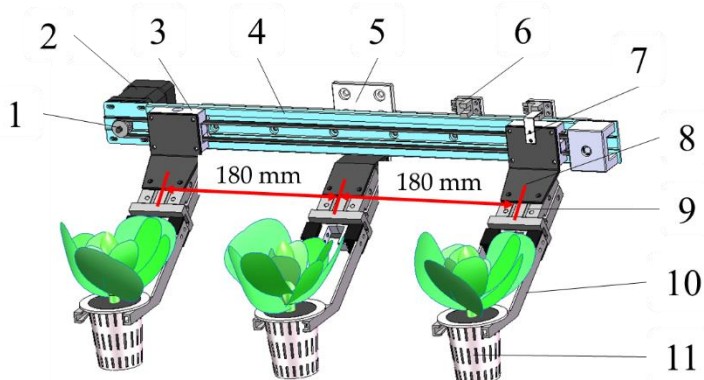

**Figure 15.** Three-claw transplanting actuator: 1. Pulley. 2. Stepper motor. 3. Slider. 4. Guide rail. 5. Main board. 6. Sensor. 7. Induction sheet. 8. Fixing plate. 9. Pneumatic finger. 10. Gripper. 11. Transplanting cup.

The operation plan of the three-claw transplanting actuator is shown in Figure 16. A detection and supplementation station is set between the seedling tray and the transplanting plate. First, the transplanting actuator grabs the three transplanting cups in the seedling tray and transports them to the detection station. If the three seedlings are all qualified, the actuator puts the seedlings into the transplanting plate. If there are unqualified seedlings or empty cells in the three seedlings, discard the unqualified seedlings or empty cells at the detection station, then the actuator moves to the seedling replenishment station, and the qualified seedlings are supplemented to the transplanting actuator. Finally, three seedlings were placed on the transplanting plate, and repeated the cycle.

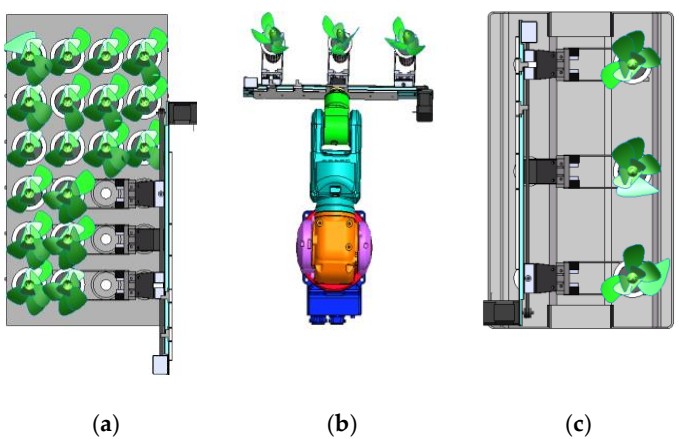

(**a**)　　　　　　　　　　(**b**)　　　　　　　　　　(**c**)

**Figure 16.** Three-claw transplanting actuator work plan. (**a**) Taking seedlings; (**b**) detection; (**c**) transplanting.

The analysis shows that the width of lettuce seedlings at 22 days of seedling age is about 110 mm, the spacing between transplanting seedlings is 180 mm, and there is no overlap of leaves between seedlings. Compared with the single-claw transplanting robot, the operation efficiency is greatly improved. The transplanting cup is lifted 20 mm before grasping, and the single-claw actuator can grasp the transplanting cup smoothly. The success rate of placing the transplanting cup into the transplanting plate depends on the running speed of the robot. Through the test, when the running speed is 50%, the efficiency is 248.3 plants/h, and the operation is stable. The three-claw actuator can grab three seedlings each time. We estimate that its operation efficiency is about 3 times that of the single-claw actuator. The detailed data need to be tested further. Therefore, the vision detection method of leafy vegetable seedlings and the design scheme of the transplanting actuator

proposed in this paper provide a theoretical basis for the subsequent development of efficient transplanting robots.

## 4. Conclusions

In response to the automatic detection of seedling empty cells and unqualified seedlings by leafy vegetable transplanting robots, a seedling detection method based on vision technology was proposed, and image recognition and detection technology was studied. The 17 d, 20 d, and 22 d lettuce seedlings were used as the object. As the seedling age increases, the probability of leaf crossing between seedlings increases, and the detection accuracy gradually decreases.

From the perspective of detection accuracy, 17 d seedlings had the highest detection accuracy, and the leaves between seedlings were almost free of borders, which is suitable for mechanical transplanting operations. From the perspective of transplanting agronomy, the 22-day-old seedlings grew 5–6 leaves, and the growth reached the requirements of transplanting. However, the occlusion of leaves between seedlings was serious and the detection accuracy rate was decreased. In order to solve the problem of influence of leaf occlusion between seedlings on detection accuracy, a seedling cup grasping–detection–transplanting operation scheme was proposed.

According to the transplanting agronomic requirements and the specifications of the seedling tray and the transplanting plate, lettuce with a seedling age of 22 days was selected as the standard for transplanting, and it was proposed to test the seedlings during the transplanting process to prevent the leaves from blocking each other and crossing the border between the seedlings, so as to accurately identify growth indicators such as leaf area pixels. Therefore, the transplanting age of lettuce seedlings can be selected as 22 d.

This paper aims to determine the detection accuracy of qualified seedlings at different seedling ages by the visual system, and the detection accuracy of the whole tray is low. The design schemes of single-claw and multi-claw transplanting actuators were discussed, and it was concluded that the three-claw transplanting actuator had advantages in terms of operation efficiency. We adopt the operation of taking first and then detecting, and avoiding the problem of the detection accuracy being affected by the shelter. The above research results can provide theoretical basis and design reference for the research and development of transplanting robot vision system and transplanting actuator.

**Author Contributions:** Conceptualization, W.F., J.G., and K.J.; methodology, W.F. and C.Z.; software, J.G. and K.J.; validation, W.Z. and Y.T.; formal analysis, W.F. and K.J.; investigation, J.G. and C.Z.; data curation, W.F., J.G., and K.J.; writing—original draft preparation, J.G. and K.J.; writing—review and editing, W.F. and W.Z.; supervision, W.Z.; funding acquisition, C.Z. All authors have read and agreed to the published version of the manuscript.

**Funding:** This research was supported by the integration and demonstration in unmanned farming technology of Tianjin Intelligent Agriculture Research Academy (01 × 0000138), the Key Research and Development Program of Ningxia China (2019BBF02010), the construction of Beijing engineering laboratory of agricultural Internet of Things technology (PT2022-27), the BAAFS Innovation Ability Project (KJCX20220403).

**Institutional Review Board Statement:** Not applicable.

**Informed Consent Statement:** Not applicable.

**Data Availability Statement:** The data presented in this study are available upon demand from the correspondence author at (jiangk@nercita.org.cn).

**Conflicts of Interest:** The authors declare no conflict of interest.

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
