# Peer review of "Detection Method and Experimental Research of Leafy Vegetable Seedlings Transplanting Based on a Machine Vision"

_agronomy, doi:10.3390/agronomy12112899_

Round 1

Reviewer 1 Report

The authors presents the mechanical hydroponic seeding transplanting solutions for China with interesting costs and sustainability. The evaluation method to different seedling ages based on image processing for leaf area measurement. It could be valuable only to substantiate the technological solution (proposed transplanting scheme (Figure 16)). However, neither in manuscript itself, nor the conclusions develops proposed image processing method. The authors never described how they collected data and then did not really analyze the data that they did include. The results and discussion chapter is roughly. I think the core concept of this paper is to evaluate and test the performance of image processing in lettuce seedling with different seedling age. It is still at its initial stage which is more like a report or conference paper. There seems to be much work to be done for this study.

Introduction-

The authors review several seedling detection method based on image processing technology. The highlight of proposed image processing method among published methods should be clarify. Some methods were explained and other non. Please, explain in the conclusion section if your method is safety, robust, decentralized or will be especially apparent with further implementation of lettuce vegetable transplantation robot technology.

Materials and Methods-

1.      I am not quite sure how finding out the relationship between seedling age (17, 20, 22 day after seedlings) and transplanting age for optimal transplanting operation plan will benefit the subsequent thinning operation of the transplanting robot with visual system. Basically, image processing can be used to determine which seedlings need to be transplanted. This also involves how the subsequent transplant machine can optimally perform the transplant to reduce the transplant time (transplanting operation plan). The author should analyze in more depth under which conditions the transplanter can quickly thin or remove the seedlings. Detection of leaf area alone is insufficient to determine seedling quality. Basically, if the seedlings are of the same variety, there will be no significant difference in their growth. In addition, optimal thinning conditions need to be defined.

2.      In addition, it is recommended that some uncertain factors should be considered in this detection method, including green moss on the plug tray, lettuce of different varieties or colors, unstable light intensity, etc., to verify the robustness and flexibility of the proposed image processing method. What does Figure 10 represent? The content is unclear. Please clarify, especially what does the red box and the green box represent?

Results and Discussions-

1.      Please be careful to present the results clearly, and ensure that deductions and interpretations are provided in a well-organized manner. For example, content about sparse transplanting operation (line 379-381) and referring to Fig. 13 seems to be drawing conclusions about low transplanting efficiency. I cannot see the specific statistical difference results.  

2.      On lines 459-461, you say “The design schemes of single-claw and multi-claw transplanting….. and detection accuracy.” I cannot see on what this claim is based.

3.      There would need to be some clear demonstration that your implementation of what may be a detection model for lettuce does do more than “a theoretical basis for the subsequent development of efficient transplanting robots (line 436-440)”.

4.      To properly justify the outcomes of the results and illustrate the merits of the paper, it is suggested that more discussion should be spent describing the innovation of proposed “machine vision” as compared to the existing/conventional robot or vision tracking platform for detection and transplanting items.

5.      The content of the paper lacks the explanation of the test results of the image collection under different seeding age settings and cross border condition.

Others-

The format of literature review is not consistent, especially the uppercase and lowercase of the paper title. These are a selection, there are loads more, please check again.

Author Response

Dear Editors and Reviewers,

    On behalf of all the authors, I would like to sincerely appreciate your valuable comments on the manuscript. Your comments not only provide constructive suggestions on improving the quality of the manuscript, but also lead us to in-depth thinking of our approaches. We will benefit from them for our future research. Based on your review comments, we have revised the manuscript accordingly and highlighted the changes. In the following, we described the changes we made corresponding to each comment.

    The authors presents the mechanical hydroponic seeding transplanting solutions for China with interesting costs and sustainability. The evaluation method to different seedling ages based on image processing for leaf area measurement. It could be valuable only to substantiate the technological solution (proposed transplanting scheme (Figure 16)).However, neither in manuscript itself, nor the conclusions develops proposed image processing method. The authors never described how they collected data and then did not really analyze the data that they did include. The results and discussion chapter is roughly. I think the core concept of this paper is to evaluate and test the performance of image processing in lettuce seedling with different seedling age. It is still at its initial stage which is more like a report or conference paper. There seems to be much work to be done for this study. 

Authors’ Response: We really appreciate your positive and constructive comments on our manuscript. The manuscript was revised carefully based on the comments. We developed the recognition and single-claw transplanting robot for leafy vegetable seedlings. The vision system can judge empty cell and unqualified seedlings, and realize the operation process of the transplanting robot such as taking seedlings trays, judging qualified seedlings, and transplanting. In addition, by testing the images of tray seedlings at different seedling ages, the recognition accuracy of different age seedlings is determined, and the design scheme and operation efficiency of single-claw and three-claw transplanting actuators is compared. The experimental results of the whole machine will be sorted and published in the future.

Introduction-

   The authors review several seedling detection method based on image processing technology. The highlight of proposed image processing method among published methods should be clarify. Some methods were explained and other non. Please, explain in the conclusion section if your method is safety, robust, decentralized or will be especially apparent with further implementation of lettuce vegetable transplantation robot technology.

Authors’ Response: In this paper, we obtained the pixel value of the leaf area of the seedling based on image recognition technology, determined the recognition accuracy of qualified seedlings at different seedling age, and proposed a single-claw transplanting scheme with first taking and then recognition. In addition, the structure and principle of three-claw transplanting actuator was analyzed, which provided the possibility to improve the transplanting efficiency.

 Materials and Methods-

1. I am not quite sure how finding out the relationship between seedling age (17, 20, 22 day after seedlings) and transplanting age for optimal transplanting operation plan will benefit the subsequent thinning operation of the transplanting robot with visual system. Basically, image processing can be used to determine which seedlings need to be transplanted. This also involves how the subsequent transplant machine can optimally perform the transplant to reduce the transplant time (transplanting operation plan).The author should analyze in more depth under which conditions the transplanter can quickly thin or remove the seedlings. Detection of leaf area alone is insufficient to determine seedling quality. Basically, if the seedlings are of the same variety, there will be no significant difference in their growth. In addition, optimal thinning conditions need to be defined. 

    Authors’ Response: When the seedling age is 17 days, the number of leaves is 3-4, and the leaves don’t cross the boundary. Mechanical transplanting can be carried out, but the cultivation period is long. When the seedling age is 20 days, the leaves cross the boundary less. When the seedling age is 22 days, the number of leaves is 5-6, and the leaves cross the boundary more serious. The root structure of seedlings with 22 days has been established, which can adapt to the growth environment of nutrient solution faster. The transplanting process of single-claw actuator is taking seedlings, recognition and transplanting, and the transplanting efficiency is lower. After the three-claw actuator takes the seedlings, the space of the three seedlings becomes larger, and there is no shelter between the leaves. The vision system can accurately identify the pixel value of the seedling leaf area, which can improve the transplanting efficiency. With the detection index of qualified seedlings by leaf area of seedlings, the visual system has fast detection speed, simple structure, and can quickly identify empty cell and weak seedlings. The qualified seedlings standard is set, which provides a judgment basis for the transplanting actuator. Unqualified seedlings are removed, and qualified seedlings are replenished for the transplanting actuator through the seedling replenishing device to achieve rapid transplant. Visual inspection scheme can also be formulated according to different varieties.

2. In addition, it is recommended that some uncertain factors should be considered in this detection method, including green moss on the plug tray, lettuce of different varieties or colors, unstable light intensity, etc., to verify the robustness and flexibility of the proposed image processing method. What does Figure 10 represent? The content is unclear. Please clarify, especially what does the red box and the green box represent? 

    Authors’ Response: The seedling age is 22 days, and moss does not grow in the plug tray. When the color of the seedling changes, the leaves can be separated from the background perfectly by modifying parameters such as image grayscale and binaryzation. In this paper, we have completed the seedling image acquisition at a fixed time, and the light intensity is relatively stable. Next, we will acquire the seedling image for the whole time to improve the robustness of the visual system. Figure 10 shows the detection results of leaf area of seedling age for 17d, 20d and 22d. The red box indicates unqualified seedlings and the green box indicates qualified seedlings. Please see lines 289-290.

Results and Discussions-

1. Please be careful to present the results clearly, and ensure that deductions and interpretations are provided in a well-organized manner. For example, content about sparse transplanting operation (line 379-381) and referring to Fig. 13 seems to be drawing conclusions about low transplanting efficiency. I cannot see the specific statistical difference results.  

    Authors’ Response: The following picture is our developing single-claw transplanting robot. According to the test, the operation efficiency is 248.3 plants / h. We believe that the efficiency should be further improved, so we propose a three-claw transplanting scheme, and the test data will be published in the future.

2. On lines 459-461, you say “The design schemes of single-claw and multi-claw transplanting….. and detection accuracy.” I cannot see on what this claim is based. 

    Authors’ Response: In fact, the detection accuracy is the same in both cases, and the discussion on detection accuracy is deleted in line 469.

3. There would need to be some clear demonstration that your implementation of what may be a detection model for lettuce does do more than “a theoretical basis for the subsequent development of efficient transplanting robots (line 436-440)”.

    Authors’ Response: The model detection for Leaf vegetable seedlings can quickly detect empty and weak seedlings, and the recognition accuracy for the elder seedling leaves is reduced. Therefore, a single-claw actuator is proposed to take seedlings first and then detect, which the recognition accuracy is not interfered by the leaves shelter, and the recognition accuracy is improved. In addition, the transplanting process of single-claw actuator includes taking seedling, detecting and transplanting. Only one seedling is transplanted each time, and this efficiency is low. According to the structure of the transplant plate, a design scheme of the three-claw transplanting actuator is proposed. The studies mentioned above provides a basis for the research and development of the transplanting robot.

4. To properly justify the outcomes of the results and illustrate the merits of the paper, it is suggested that more discussion should be spent describing the innovation of proposed “machine vision” as compared to the existing/conventional robot or vision tracking platform for detection and transplanting items. 

    Authors’ Response: According to your suggestions, we will further improve and improve the algorithm innovation of machine vision. However, it will take more time to modify the visual system of test prototype. Please bear with us. At present, the problem that the shelter of elder seedling leaves affects the recognition accuracy has not been solved. This paper aims to identify the growth state of tray seedlings by the visual system, provide a basis for the design of the transplanting actuator, propose to take seedlings before detection, avoid the shelter of leaves, which provide a reference for the research of the transplanting robot.

6. The content of the paper lacks the explanation of the test results of the image collection under different seeding age settings and cross border condition. 

    Authors’ Response: In Section 3.2.1, we discussed the influence of different seedling ages and leaf boundary crossing conditions on the recognition accuracy of qualified seedlings. Please see lines 330-379.

Others-

    The format of literature review is not consistent, especially the uppercase and lowercase of the paper title. These are a selection, there are loads more, please check again.

    Authors’ Response: According to your comments, we have revised the title of the paper and the format of literature review. Please lines 1-3、78、85.

Reviewer 2 Report

Overall, the manuscript is correctly written. The current manuscript is interesting and novel, so due to that the manuscript could be valuable contribution to the scientific community. Generally, the study appear to be sound, well-designed and could be also of practical value. Therefore, the undertaken issue is interesting and worth of disseminating. However, I suggest the authors add a few additions and corrections to this paper:

The introduction is well written with a large number of citations. However, it seems extremely long with a large amount of general information that does not contribute significantly to this manuscript and hypothesis. Thus, I would advise to reduce the introduction by 20-30% to an acceptable size.

Very little information is given about the effect of the color and/or shape of the seeding tray and transplanting cap on the accuracy of the presented method. For the lines 171-178, whether a lot of experimentation will be needed again if other transplant caps are used? For the lines 237-249, Are the specifications of the seedling tray typical or non-standard?

Please specify the parameters for the seeding tray – for the line 241 presented 28-hole seedling tray and for the line 378 presented 24-hole seedling tray. Shouldn't they be the same?

Please elaborate on your choice of the dividing lines - why 2500, 4500 and 4800 pixels?

For the Table 2, this would be great if you can add some numerical values to the "cross-border leaves" column.

Inside the discussion section, I did not find any comparison with potential analogues, if possible, add an indirect comparison with systems that do the same job.

Author Response

Dear Editors and Reviewers,

    On behalf of all the authors, I would like to sincerely appreciate your valuable comments on the manuscript. Your comments not only provide constructive suggestions on improving the quality of the manuscript, but also lead us to in-depth thinking of our approaches. We will benefit from them for our future research. Based on your review comments, we have revised the manuscript accordingly and highlighted the changes. In the following, we described the changes we made corresponding to each comment.

   Overall, the manuscript is correctly written. The current manuscript is interesting and novel, so due to that the manuscript could be valuable contribution to the scientific community. Generally, the study appear to be sound, well-designed and could be also of practical value. Therefore, the undertaken issue is interesting and worth of disseminating. However, I suggest the authors add a few additions and corrections to this paper:

    Authors’ Response: We really appreciate your positive and constructive comments on our manuscript. The manuscript was revised carefully based on the comments.

1. The introduction is well written with a large number of citations. However, it seems extremely long with a large amount of general information that does not contribute significantly to this manuscript and hypothesis. Thus, I would advise to reduce the introduction by 20-30% to an acceptable size.

    Authors’ Response: According to your opinion, we have deleted similar documents in the introduction. Please see lines 71、75、77-80、83、85、89、93、95.

2. Very little information is given about the effect of the color and/or shape of the seeding tray and transplanting cap on the accuracy of the presented method. For the lines 171-178, whether a lot of experimentation will be needed again if other transplant caps are used? For the lines 237-249, Are the specifications of the seedling tray typical or non-standard?

    Authors’ Response: This vision system is not affected by the shape of the transplanting cup, but there need be color difference between the leaf and the background. The materials of seedling tray and transplanting cup are ABS and PP respectively, and both are white. It is unnecessary to conduct more tests again for other transplanting cups. In addition, the specification of seedling tray is non-standard customized, suitable for lettuce seedling.

3. Please specify the parameters for the seeding tray – for the line 241 presented 28-hole seedling tray and for the line 378 presented 24-hole seedling tray. Shouldn't they be the same?

    Authors’ Response: For single-claw transplanting actuator, the seedling tray is 28 cells, because the operation of single-claw actuator was not affected by the number of cell. However, For the three claw actuator, 3 seedlings are transplanted each time, so there will be one line left when the 28 cells tray is operated. Therefore, the tray is changed to 24 cells, which is more suitable for the three-claw actuator operation.

4. Please elaborate on your choice of the dividing lines - why 2500, 4500 and 4800 pixels?

    Authors’ Response: We tested the leaf area pixels of qualified seedlings at different seedling ages to determine the dividing line of different seedling ages. These data need to be demarcated for different varieties of seedlings, because the shape and color of seedlings have changed.

5. For the Table 2, this would be great if you can add some numerical values to the "cross-border leaves" column.

    Authors’ Response: According to your opinion, we have supplemented the data in Table 2. Please see the table 2.

6. Inside the discussion section, I did not find any comparison with potential analogues, if possible, add an indirect comparison with systems that do the same job.

    Authors’ Response: In the introduction, the recognition accuracy of similar vision systems is introduced, and the test indicators include empty cell, unqualified seedlings and qualified seedlings. In addition, we tested seedlings with different seedling ages, while others study tested seedlings with a certain seedling age. Due to different test varieties and conditions, the results were quite different, so we did not compare with each other. Please bear with us.

Reviewer 3 Report

The authors reported the development of a computer vision system for seedling transplants. The proposed system detected different growth parameters to identify the quality of lettuce seedlings at different ages. The reported accuracy is about to 86% for seedling detection. Since transplanting and sorting healthy seedlings is a labor-intensive task, this study is critical to address the challenges leafy green growers face. The authors have explained the methodology and provided fair results and discussion in the manuscript. However, I have some serious concerns about the scope of this paper; my comments are below:

1.      The article lacks clarity in the scope. The article talks about developing a machine vision system for seedling transplants, but the later part of the paper discusses the design of the robot. However, no design methodology is presented. Did the author also develop the robotic system, or was the study conducted in the simulation? The authors must revise the paper.

2.      Line 116-118: The authors talked about the working process, and this information in the materials and methods section assuming the whole system is physically available and developed during the study. What were the position errors for the robot and conveyors? Were the conveyors synchronized with the robot's motion?

3.      Working in simulation is very different than in actual conditions. The authors claimed the operation efficiency is three times of a three-claw robotic system; how did the authors reach this conclusion? How many times did the claw fail to pick a transplant cup, and how many failures for placement in the transplant tray, either for a single claw or three claws? Did the authors perform any tests for this? If no tests are done, the claims should not be made on efficiency or accuracy.

4.      The methodology presented for the machine vision system is trivial. There is tremendous research available for implementing classical image processing, so I do not think providing details on image processing is necessary. It may be good for the thesis or dissertation, but it should be reduced to an article.

5.      How the information in Table 1 and Fig 11 are different from each other. Why did the author choose to include a table and similar information in the form of a figure? Isn't it the repetition of the same information?

6.      The conclusion section should be revised. It should only highlight what was concluded from the study, not repeat the results.

Author Response

Dear Editors and Reviewers,

    On behalf of all the authors, I would like to sincerely appreciate your valuable comments on the manuscript. Your comments not only provide constructive suggestions on improving the quality of the manuscript, but also lead us to in-depth thinking of our approaches. We will benefit from them for our future research. Based on your review comments, we have revised the manuscript accordingly and highlighted the changes. In the following, we described the changes we made corresponding to each comment.

    The authors reported the development of a computer vision system for seedling transplants. The proposed system detected different growth parameters to identify the quality of lettuce seedlings at different ages. The reported accuracy is about to 86% for seedling detection. Since transplanting and sorting healthy seedlings is a labor-intensive task, this study is critical to address the challenges leafy green growers face. The authors have explained the methodology and provided fair results and discussion in the manuscript. However, I have some serious concerns about the scope of this paper; my comments are below:

    Authors’ Response: We really appreciate your positive and constructive comments on our manuscript. The manuscript was revised carefully based on the comments.

1. The article lacks clarity in the scope. The article talks about developing a machine vision system for seedling transplants, but the later part of the paper discusses the design of the robot. However, no design methodology is presented. Did the author also develop the robotic system, or was the study conducted in the simulation? The authors must revise the paper. 

    Authors’ Response: The following figure shows the prototype of single-claw transplanting robot developed by us. After testing, the working efficiency is 248.3 plants / h. The test data will be published in the future. This paper aims to determine the detection accuracy of qualified seedlings at different seedling ages by the visual system, and the detection accuracy of the whole tray is low. We adopt the operation that taking first and then detecting, discuss the design schemes of single-claw and three-claw actuators, and avoid the problem that the detection accuracy is affected by the shelter.

2. Line 116-118: The authors talked about the working process, and this information in the materials and methods section assuming the whole system is physically available and developed during the study. What were the position errors for the robot and conveyors? Were the conveyors synchronized with the robot's motion?

    Authors’ Response: The position relationship between the robot and the two conveyor belts is to taking and transplanting seedlings. The photoelectric sensor is used to detect the signals of seedling tray and transplant board, and then the mechanical parts are used for secondary positioning. In the robot control system, the space coordinates of taking, detection and transplanting seedlings are set. The position error of robot movement is ± 0.02mm.

3. Working in simulation is very different than in actual conditions. The authors claimed the operation efficiency is three times of a three-claw robotic system; how did the authors reach this conclusion? How many times did the claw fail to pick a transplant cup, and how many failures for placement in the transplant tray, either for a single claw or three claws? Did the authors perform any tests for this? If no tests are done, the claims should not be made on efficiency or accuracy. 

    Authors’ Response: The transplanting cup is lifted 20mm before grasping, and the single-claw actuator can grasp the transplanting cup smoothly. The success rate of placing the transplanting cup into the transplanting plate depends on the running speed of the robot. Through the test, when the running speed is 50%, the efficiency is 248.3 plants / h, and the operation is stable. The three-claw actuator can grab three seedlings each time. We estimate that its operation efficiency is about 3 times that of the single-claw actuator. The detailed data needs to be tested further. Please see lines 441-442.

4. The methodology presented for the machine vision system is trivial. There is tremendous research available for implementing classical image processing, so I do not think providing details on image processing is necessary. It may be good for the thesis or dissertation, but it should be reduced to an article. 

    Authors’ Response: I agree with your view that there is tremendous research available for implementing classical image processing. I also think that the test conditions and processing methods are not always the same. If the image processing process is simplified, it is not conducive for readers to understand and reproduce the test results.

5. How the information in Table 1 and Fig 11 are different from each other. Why did the author choose to include a table and similar information in the form of a figure? Isn't it the repetition of the same information?

    Authors’ Response: Table 1 show the test results of leaf area of seedlings age at 17d, and Figure 11 show the statistical results at 17d, 20d and 22d. We think can be expressed and compared more clearly.

6. The conclusion section should be revised. It should only highlight what was concluded from the study, not repeat the results.

    Authors’ Response: According to your opinion, we have revised the conclusion. Please see lines 449-450、452-453、456-461.

Round 2

Reviewer 3 Report

The authors have revised the manuscript and provided reasonable responses to the comments. I have a few minor comments for the author’s consideration.

The data and figures provided in the responses should be added to the manuscript. If the figure or data is not within the scope of this study, then please don’t share them for the review process. Justify your work based on what you have presented in the manuscript. The reviewer may be satisfied with the additional material/figure. Still, the reader may have similar questions after publishing, and the reader will not have access to the extra material you provided for the review process. Please add all material or figures you added to the response file. I strongly recommend that for all the comments where you provided the justification/explanation/new data about a certain point, add it to the manuscript file. A suggestion: In the responses, please mention the line numbers to reflect any changes/edits done in the manuscript for particular comments, if any. 

Author Response

November 17, 2022

Dear Editors and Reviewers,

    On behalf of all the authors, I would like to sincerely appreciate your valuable comments on the manuscript. Your comments not only provide constructive suggestions on improving the quality of the manuscript, but also lead us to in-depth thinking of our approaches. We will benefit from them for our future research. Based on your review comments, we have revised the manuscript accordingly and highlighted the changes. In the following, we described the changes we made corresponding to each comment.

        The authors have revised the manuscript and provided reasonable responses to the comments. I have a few minor comments for the author’s consideration.

        Authors’ Response: We really appreciate your positive and constructive comments on our manuscript. The manuscript was revised carefully based on the comments.

    The data and figures provided in the responses should be added to the manuscript. If the figure or data is not within the scope of this study, then please don’t share them for the review process. Justify your work based on what you have presented in the manuscript. The reviewer may be satisfied with the additional material/figure. Still, the reader may have similar questions after publishing, and the reader will not have access to the extra material you provided for the review process. Please add all material or figures you added to the response file. I strongly recommend that for all the comments where you provided the justification/explanation/new data about a certain point, add it to the manuscript file. A suggestion: In the responses, please mention the line numbers to reflect any changes/edits done in the manuscript for particular comments, if any. 

        Authors’ Response: We are really grateful for your precious advice. According to your comment, we have supplemented the relevant materials, and data in the manuscript, which make the paper more accurate and clarify. For details, see lines 114-118, 447-453, 471-474, 486-493.
